# Odontogenic cysts and tumors detection in panoramic radiographs using Deep Convolutional Neural Network(DCNN)

**Tae-Hoon Yong**[1]                                                    LOUISYONG9512@GMAIL.COM

**Sang-Jeong Lee** [2]                                                    SJLEE89@SNU.AC.KR

**Won-Jin Yi** [3]                                                    WJYI@SNU.AC.KR

[1] *Department of Computer Engineering, School of Engineering, Hongik University, Seoul, Korea*

[2] *Department of Biomedical Radiation Sciences, Graduate School of Convergence Science and Technology, Seoul National University, Seoul, Korea*

[3] *Seoul National University Dental hospital, Department of Oral and Maxillofacial Radiology, Seoul, South Korea*

## Abstract

Diseases that require surgery, such as cysts or tumors that occur in the oral maxillofacial region, have been often missed or misdiagnosed despite the importance of early detection. Computer-assisted diagnostics using a deep convolution neural network (DCNN), a machine learning technology based on artificial neural networks, can provide more accurate and faster results. In this study, we will investigate a method for automatically detecting five diseases that frequently occur in the oral maxillofacial region using DCNN in panoramic radiographs.

**Keywords:** Object Detection, Region Proposal Networks, Dental panoramic images

## 1. Introduction

A lot of studies are currently being conducted on the diagnosis using deep learning, and it is actively used in the medical field. In particular, the technique of predicting the region of a disease in a radiological image conduct a great help to a radiologist. Among other things, the deep learning technology to detect objects with CNN (Krizhevsky et al., 2012) backbone such as Region proposals with Convolutional Neural Network (RCNN) (Girshick et al., 2014), Fast RCNN (Girshick, 2015) and Faster RCNN (Ren et al., 2015) is a major innovation in the existing medical system. Recently, a new technology has been announced that has improved performance called YOLO (Redmon et al., 2016). YOLO has the advantage of faster image analysis than Faster RCNN. In this paper, we will perform detecting cyst and tumor diseases using YOLO-V3 networks and compare the results.

## 2. Proposed method

### 2.1. Retrospective collection of the train and test datasets

A total of 1182 panoramic images of the dental clinic of Seoul National University from 1999 to 2017 were analyzed. There are images with 291 Odontogenic Cyst(OKC), 285 Dentigerous Cyst(DC), 272 Osteomyelitis(OM), 145 Periapical Cyst(PC) and 230 Ameloblastoma(AB). The panoramic image was based on data confirmed to be the same lesion histopathologically as dental radiographic findings.

### 2.2. Labeling of the datasets

Each image was labeled with an image with or without OKC, DC, OM, PC and AB. For annotation, we labeled each image using YOLO-mark and the work was done by radiologists.

### 2.3. Data augmentation

Augmentation was performed with rotation, horizontal flip and gamma correction. A rotation angle from -1 ° to 1 ° in 0.5 ° increments and a gamma range from 0.7 to 1.3 in increments of 0.3 ° were used for the augmentation. So the total number of images has increased by 30 times the number of existing images, and the learning time has increased even more. This not only makes learning better, but it also gives better performance in various test images.

### 2.4. Network structure

DCNN-based neural network was used to detect odontogenic cysts and tumor in dental panoramic images. The neural networks used modified YOLO-v3 to detect odontogenic cysts and tumor, which uses DarkNet-53. DarkNet-53 we used as a backbone was pretrained on ImageNet. To improve learning performance, the resolution of the image was raised from the previous 256 by 256 to 608 by 608. In addition, we used a learning rate of 0.001, decay of 5 x $10^{-4}$ , momentum of 0.9 and batch of 64. All experiments were performed using four GPUs (NVIDIA 1080ti with 11GB memory).

## 3. Results

When using the panoramic image as input, Figure 1 shows the location of the detected diseases as well as the name of the disease. We used mean average precision (mAP), precision, and recall as indicators for accuracy evaluation. The mAP is mainly used to evaluate the performance of object detection algorithms with indices using precision and recall. Table 1 showed the results of detecting cysts and tumor in panoramic images. The results of detecting Odontogenic Cyst showed an average precision of 90.64%, a precision of 0.98, and a recall rate of 0.86. Second, detecting Dentigerous Cyst recorded 90.91%, 1.00, and 0.93, respectively. Third, Osteomyelitis results were 99.48%, 0.97, 0.97. Fourth, the results were 89.67%, 0.98, and 0.87 in the Periapical Cyst. Finally, for ameloblastoma, values of 100.00%, 1.00, and 1.00 were obtained.

**Ground Truth**         **YOLO-V3 detection**

Figure 1: Example of detecting OKC and OM in odontogenic cysts and tumors. The images on the left are images of the input images and the bounding boxes labeled by a radiologist, and the images on the right are results of YOLO v3 detecting the same input images.

Table 1: The results of mAP, precision and recall using YOLO-V3

|  | **AP(%)** | **precision** | **recall** |
|---|---|---|---|
| OKC | 90.64 | 0.98 | 0.86 |
| DC | 90.91 | 1.00 | 0.93 |
| OM | 99.48 | 0.97 | 0.97 |
| PC | 89.67 | 0.98 | 0.87 |
| AB | 100.00 | 1.00 | 1.00 |
| **Mean** | 94.14 | 0.99 | 0.93 |

## 4. Conclusion

In this study, we have developed a method to automatically detect Odontogenic Cyst, Dentigerous Cyst, Osteomyelitis, Periapical Cyst and Ameloblastoma in panoramic radiographs using DCNN technique. Our study method showed high accuracy despite limited clinical data. Also, disease detection using these object detection techniques has been used in many medical fields (Liu et al., 2017; Sa et al., 2017), but the dental field has not yet achieved such great results. We will conduct research that makes a great contribution to this field.

## Acknowledgments

This work was supported by the Technology Innovation Program (10063389) funded by the MOTIE, Korea.

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
