# OpenReview forum: "Odontogenic cysts and tumors detection in panoramic radiographs using Deep Convolutional Neural Network(DCNN)"
_MIDL.io/2019/Conference/Abstract — MIDL Abstract 2019_

### Official Review · AnonReviewer1 · 2019-04-29
**Dental cyst and tumor detection using YOLO object detection network**

**Rating:** 2
**Confidence:** 2

**Review:**

Summary: Authors trained YOLOv3 network to detect 5 different pathologies (cyst/tumor) in panoramic radiographs.

Strengths:
Authors claim to be the first to achieve high results in dental field thanks to the employed object detection model.

Weaknesses:
There is no reference to other work in order to comprehend the significance of the achieved quantitative results. It is not clear how and if the dataset was split between training and test. The manuscript does not propose a novelty, but apply an existing approach to dental field.

Major concerns:
Out of the 1182 images in the dataset, authors did not indicate the split for training/validation/test. This is concerning, as it makes me think that the presented results are perhaps on the training set.

Sec 2.4. Please clarify how does increasing image resolution improve the learning performance. Do images look more similar to natural images from ImageNet after such rescaling?

Comments:
Sec 2.2: In sounds as if GT consists of image masks, and not 4 scalars for each instance of tumor/cyst. Please clarify.
Sec 2.2: Citation needed for YOLO-mark.
Sec 2.3 “learning time” is ambiguous. Please clarify if authors meant to say number of training iterations.

---

### Official Review · AnonReviewer2 · 2019-04-30
**Nice application of state-of-the-art object detector networks**

**Rating:** 3
**Confidence:** 1

**Review:**

The authors evaluate the performance of a popular object detector network (YOLO-V3) for the detection of odontogenic cysts and tumors detection in panoramic radiographs. This work is a good example of best practices in machine-learning based deep learning albeit using an off-the-shelf object detector network. The application to dental x-ray images seems novel as this field has not been explored much by medical imaging researchers. The dataset is large enough to make the results seem very promising for this application.

---

### Decision · Program_Chairs · 2019-05-06
**Acceptance Decision**

Accept